# 🐾 Pack of LLMs: Model Fusion at Test-Time via Perplexity Optimization

**Costas Mavromatis**
University of Minnesota
mavro016@umn.edu

**Petros Karypis**
University of California San Diego
pkarypis@ucsd.edu

**George Karypis**
University of Minnesota
karypis@umn.edu

## Abstract

Fusing knowledge from multiple Large Language Models (LLMs) can combine their diverse strengths to achieve improved performance on a given task. However, current fusion approaches either rely on learning-based fusers that do not generalize to new LLMs, or do not take into account how well each LLM understands the input. In this work, we study LLM fusion at *test-time*, which enables leveraging knowledge from arbitrary user-specified LLMs during inference. We introduce *Pack of LLMs* (PackLLM), an effective method for test-time fusion that leverages each LLM's expertise, given an input prompt. PackLLM performs model fusion by solving an optimization problem for determining each LLM's importance, so that perplexity over the input prompt is minimized. First, our simple PackLLM$_{sim}$ variant validates that perplexity is a good indicator for measuring each LLM's expertise. Second, our PackLLM$_{opt}$ variant approximately solves the perplexity minimization problem via a greedy algorithm. The derived importance weights are used to combine the LLMs during inference. We conduct experiments with over 100 total LLMs on a diverse set of tasks. Experimental results show that (i) perplexity is a reliable measure for LLM fusion, (ii) PackLLM outperforms test-time fusion baselines by 1.89% accuracy points, (iii) PackLLM can leverage new LLMs to improve performance over learning-based fusion approaches by 3.92–11.94% accuracy points, (iv) PackLLM benefits over selecting the best or largest model and model merging in certain cases. Our code is provided at https://github.com/cmavro/PackLLM.

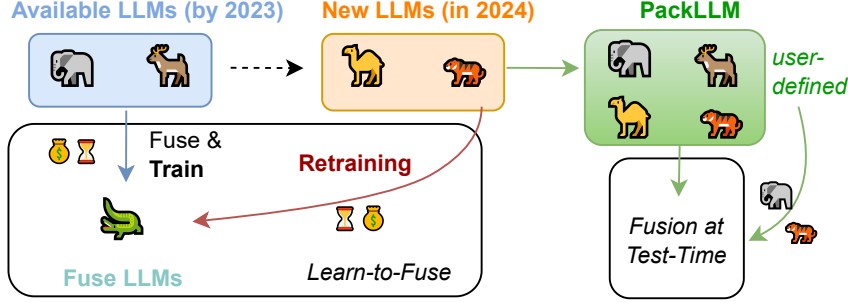

Figure 1: PackLLM performs LLM fusion at *test-time*. PackLLM (i) does not require any training of fusion models, (ii) can leverage new (more powerful) LLMs when released, and (iii) allows users to select their preferred LLMs at test-time.

# 1 Introduction

Large Language Models (LLMs) (Brown et al., 2020; Touvron et al., 2023) have achieved state-of-the-art performance on a variety of natural language processing (NLP) tasks and are beginning to be applied as Foundation Models (Bommasani et al., 2021) for a gamut of applications. As a result, we observe a sharp increase of newly released LLMs, that are pretrained on diverse corpora or fine-tuned to solve specific tasks. Moreover, it has become a strategic interest that corporations train and release their own LLMs, amplifying the number of available LLMs. LLMs usually have architectural differences as well as they use different pretraining data, and as a result, their knowledge expertise is diversified. Therefore, an emergent question is "*How can we effectively combine knowledge of the available LLMs to improve downstream performance?*".

Fusing knowledge from different models is a long-standing problem in machine learning (Ho, 1995; Sollich & Krogh, 1995; Kuncheva & Whitaker, 2003; Schapire, 2003). Regarding LLMs, current approaches train dedicated fusion modules, such as neural networks (Wang et al., 2023) or additional LLMs (Jiang et al., 2023b; Wan et al., 2024; Ding et al., 2024), that learn to combine LLMs from a seed set. However, these approaches are not *modular*, meaning they need to undergo expensive and time-consuming retraining to encompass newly released LLMs or remove specific LLMs, e.g., due to licensing issues.

We present *Pack of LLMs* (PackLLM), a *test-time* fusion method. As shown in Figure 1, PackLLM does not require any training of fusion modules, while it can combine arbitrary user-specified LLMs during inference. In order to fuse knowledge from the seed LLMs at test-time, PackLLM performs a weighted ensemble of the output logits by posing an optimization problem at test-time. PackLLM minimizes the perplexity over the input prompt, so that the fused LLM understands the task better. First, our simple PackLLM$_{sim}$ variant validates that perplexity is a good indicator for measuring each LLM's expertise. Second, our PackLLM$_{opt}$ variant approximately solves the perplexity minimization problem via a greedy algorithm. The derived importance weights are used for combining the LLMs during inference.

We conduct experiments with over 100 total LLMs and evaluate performance on language modeling tasks (Section 4) as well as on downstream tasks (Section 5). Experiments show that PackLLM's perplexity-based framework is a reliable indicator for determining fusion, outperforming approaches such as cBTM (Gururangan et al., 2023) and DExperts (Liu et al., 2021) that do not take into account how well the LLMs understand the input, while its overall performance scales better with respect to these baselines as we increase the number of expert LLMs. In addition, PackLLM achieves a significant improvement of 1.72–1.89% accuracy points (averaged over 25 tasks) over existing test-time fusion approaches, such as top expert selection and uniform ensemble. Furthermore, by employing newly released LLMs, PackLLM outperforms competing learning-based fusion approaches, such as FuseLLM (Wan et al., 2024) and FoE (Wang et al., 2023), by 3.92–11.94% accuracy points, using as little as 3.92× fewer parameters. Our contributions are summarized below:

- **Problem Formulation**: We study fusion of LLMs at test-time as a weighted ensemble and pose a perplexity minimization problem for determining the LLM importance weights. Experimental results show that PackLLM benefits over selecting the best or largest model and model merging in certain cases.

- **Algorithm**: We present PackLLM$_{opt}$ that solves the perplexity minimization problem via a greedy algorithm. We also present PackLLM$_{sim}$, a simple, but effective, perplexity-based ensemble.

- **Effectiveness**: We evaluate PackLLM on a diverse set of tasks showing that (i) perplexity is a reliable indicator for model importance, (ii) PackLLM outperforms test-time fusion baselines by 1.72–1.89% accuracy points, (iii) PackLLM can leverage newly released LLMs and outperform competing learning-based fusion approaches by 3.92–11.94% accuracy points.

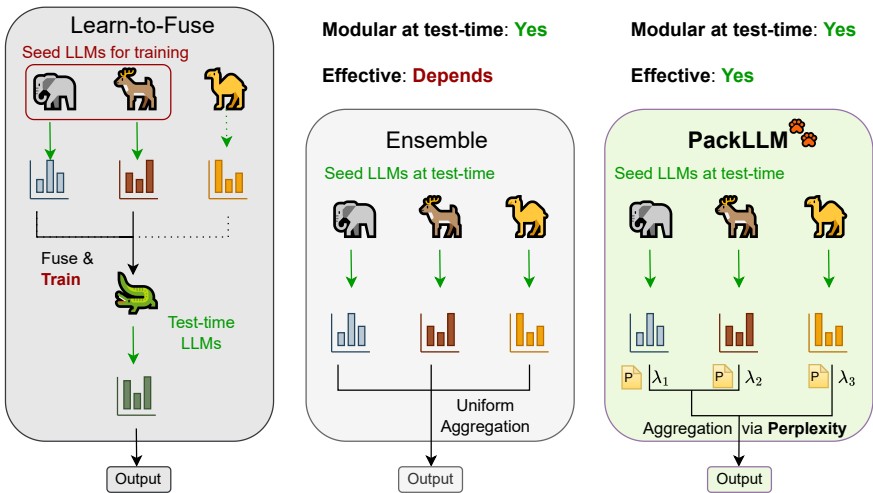

Figure 2: Overview of LLM fusion paradigms. *Left*: Learning-based fusion approaches are not modular and cannot encompass new LLMs at test-time. *Middle*: Uniform ensemble may degrade performance, given weak seed LLMs. **Right**: Our **PackLLM** approach determines the weights of the seed LLMs at test-time via perplexity to achieve effective performance.

## 2 Related Work

Regarding fusion of LLMs, *weight merging* (Yadav et al., 2023) and *mixture-of-experts* (Komatsuzaki et al., 2022) are techniques that fuse knowledge stored in LLMs by combining their model parameters. However, such approaches require that the given LLMs share the same architecture, which is limited with the current availability of powerful LLMs of varying sizes. An alternative technique is *majority voting* (Li et al., 2024a), but can only be applied to certain downstream tasks, such as classification tasks.

Recent successful fusion approaches often follow a *learn-to-fuse* framework (see Figure 2, left), where specialized learnable modules (Wang et al., 2023; Sukhbaatar et al., 2024) or even LLMs (Jiang et al., 2023b; Wan et al., 2024; Ding et al., 2024), are trained to combine knowledge from a set of seed LLMs. However, these approaches are *not modular*, meaning they cannot generalize to arbitrary user-specified LLMs and require additional retraining when models need to be added in or removed from the seed set. Approaches, such as (Feng et al., 2024), fuse the generations of different LLMs, but require a strong (large-scale) LLM as a fuser, which is computationally expensive.

On the other hand, *model ensemble* approaches (Liu et al., 2021; Gururangan et al., 2023; Liu et al., 2024; Li et al., 2024b) perform fusion at the output-level, i.e., output token logits or probabilities. However, effectively combining the LLMs during test-time, i.e., accurately determining each LLM's importance, has not been extensively studied as these approaches do not take into account how well each LLM understands the input. For instance, uniform ensemble may degrade performance if the set includes weak LLMs (Figure 2, middle).

## 3 Pack of LLMs (PackLLM) 🐾

In this work, we study the following problem:

> Given a set of user-specified LLMs $\{M_1, \ldots, M_K\}$ at test-time, how can we effectively combine the knowledge of each LLMs to achieve improved performance?

We present Pack of LLMs (PackLLM), a weighted ensemble of LLMs that fuses knowledge of arbitrary LLMs at test-time. As shown in Figure 2 (right), given a set of seed LLMs $\{M_1, \ldots, M_K\}$ at test-time, PackLLM fuses their knowledge with corresponding importance

weights $\{\lambda_1, \ldots, \lambda_K\}$. The importance weights $\lambda_k$ are calculated based on how well each LLM understands the prompt $P$.

In more detail, at each step $t$, each model $M_k$ is conditioned on the prompt $x_{<t}$, to obtain the logit score $s_k(x_t|x_{<t})$ for the next token $x_t$. Logit scores $s_k$ are unnormalized scores over the LLM's vocabulary, before applying the softmax$(\cdot)$ operation over all vocabulary tokens which transforms them into probabilities $p_{M_k}$. Here, we assume the models share the same vocabulary, but we provide a generalized version in Section 3.4. PackLLM combines output logits to obtain the final probabilities as

$$p(x_t|x_{<t}) = \text{softmax}\Big( \sum_{k=0}^{K} \underbrace{\lambda_k}_{unknown} s_k(x_t|x_{<t}) \Big), \tag{1}$$

where we assume that the weights $\lambda_k \in [0,1]$ are normalized, i.e., $\sum_{k=0}^{K} \lambda_k = 1$. In what follows, we answer the question "*How can we accurately determine weights* $\{\lambda_1, \ldots, \lambda_K\}$ *at test-time?*"

## 3.1 Prompt Perplexity Minimization

PackLLM relies on the notion that perplexity is a suitable measure for a LLM's knowledge on a certain input (where the lower the better). Perplexity is a metric that aligns with the causal language modeling objective of LLMs, enabling to measure whether a test input falls into the LLM's expertise and how the test input relates to the LLM's pretraining data (Marion et al., 2023; Gonen et al., 2022).

Given a tokenized input prompt or sequence $P = (x_0, x_1, \ldots, x_t)$, perplexity (PPL) is defined as the exponentiated average negative log-likelihood of $P$ as

$$\text{PPL}_k(P) = \exp\Big\{ -\frac{1}{t} \sum_{i}^{t} \log p_k(x_i|x_{<i}) \Big\}, \tag{2}$$

where $\log p_k(x_i|x_{<t})$ is the log-likelihood of the $i$-th token conditioned on the preceding tokens $x_{<t}$ according to model $M_k$. Note that PPL depends solely on the models and the input prompt and can be evaluated at test-time.

Leveraging perplexity as a measurement for a LLM's expertise on the test prompt, we formulate the assignments of importance weights as an optimization problem that does not require any training or annotated data and can be solved at test-time. Formally, our optimization framework is given as

$$\lambda_1^*, \ldots, \lambda_K^* = \underset{\lambda_1, \ldots, \lambda_K}{\arg\min}\Big( -\frac{1}{t} \sum_{i}^{t} \log \underbrace{\text{softmax}\Big( \sum_{k=0}^{K} \lambda_k s_k(x_i|x_{<i}) \Big)}_{p(x_i|x_{<i})} \Big),$$

$$\text{subject to } \lambda \in [0,1] \text{ and } \sum_{k}^{K} \lambda_k = 1, \tag{3}$$

where $\{\lambda_1^*, \ldots, \lambda_K^*\}$ are the optimal importance weights and $s_k$ are the output logits of each LLM. During inference, the computed weights $\{\lambda_1^*, \ldots, \lambda_K^*\}$ are used in Equation 1.

## 3.2 PackLLM$_{\text{sim}}$: Simple Perplexity-Based Weighting

An important question is whether perplexity is a reliable metric for determining fusion. As the first step, we present PackLLM$_{\text{sim}}$, a simple perplexity-based approach for determining the importance weights $\{\lambda_1, \ldots, \lambda_K\}$. Instead of solving the optimizing problem presented in Equation 3, PackLLM$_{\text{sim}}$ computes the weights $\lambda_k$ directly by using the perplexity scores over the prompt via

$$\lambda_k = \text{softmax}_k\Big( -\log \text{PPL}_k(P)/\tau \Big), \tag{4}$$

where $\tau$ is the temperature hyperparameter with default values $\tau \in \{0.1, 1\}$. Weights $\{\lambda_1, \cdots, \lambda_k\}$ are inversely proportional to the models' normalized perplexity scores, i.e., lower perplexity yields a higher weight. PackLLM$_{\text{sim}}$ serves as a validation of whether the models' perplexity scores provide useful information for determining fusion.

### 3.3 PackLLM$_{\text{opt}}$: Greedy Perplexity Optimization

Unlike the simple heuristic that PackLLM$_{\text{sim}}$ uses (Equation 4), PackLLM$_{\text{opt}}$ approximately solves the perplexity minimization problem presented in Equation 3. The problem in Equation 3 is overdetermined, where the number of equations is $t$ ($t$ prompt tokens) and the number of variables is $K$ ($K$ seed LLMs), having many solutions. Nevertheless, it can be solved at test-time via grid search (LaValle et al., 2004) by evaluating different values $\lambda_k \in [0, 1]$ with the constraint $\sum_k \lambda_k = 1$. However, exhaustive grid search is time-consuming during inference due to its combinatorial complexity on the number of LLMs.

We propose an efficient greedy algorithm, which ensembles the LLMs sequentially. The sequential nature of the algorithm reduces the optimization problem to searching between linear combinations of two models rather than the full $K$ models.

First, we compute the perplexity $\text{PPL}_k(P)$ of each LLM $k$ via Equation 2 and sort the seed LLMs $\{M_1, \ldots, M_K\}$ based on $\text{PPL}_k(P)$ as

$$[M_1^*, \ldots, M_K^*] = argsort\big(\text{PPL}_1(P), \ldots, \text{PPL}_K(P)\big), \tag{5}$$

where $[M_1^*, \ldots, M_K^*]$ are ordered by the lowest to highest perplexity scores. By using the ordered set, we can omit using irrelevant LLMs via early stopping of the sequential ensemble.

At the first step of the greedy optimization, we determine the relative weights between the top-1 and the top-2 LLMs, $M_1^*$ and $M_2^*$. We solve

$$\lambda^{(1)} = \arg\min_{\lambda} \Big( -\frac{1}{t}\sum_{i}^{t} \log \text{softmax}\big(\lambda s^{(1)}(x_i|x_{<i}) + (1-\lambda)s^{(2)}(x_i|x_{<i})\big)\Big), \tag{6}$$

where $s^{(1)}$ and $s^{(2)}$ are the output logits by $M_1^*$ and $M_2^*$, respectively, and $\lambda \in [0, 1]$. The optimization in Equation 6 is solved via grid search (greedy grid search), where different values of $\lambda \in [0, 1]$ are evaluated (the default step is 0.05). The value $\lambda^{(1)}$ that results to the lowest perplexity is used to update

$$s^{(2)} = \lambda^{(1)}s^{(1)} + (1 - \lambda^{(1)})s^{(2)} \tag{7}$$

for the next step.

The same procedure is repeated to iterate through all seed LLMs $[M_1^*, \ldots, M_K^*]$. We can perform early stopping when we find $\lambda^{(k)} = 1$, i.e., the effect of the current LLM is zero. Our overall greedy optimization is summarized in Appendix A.

### 3.4 Tokenizer Selection & Alignment

Equation 1 assumes that the logit vectors $s_k$ can be added together, which requires that the seed LLMs share the same vocabulary. However, this is not always the case, as different LLMs are trained based on different tokenizers. To address this, we employ a tokenizer selection and alignment strategy for combining LLMs with different tokenizers.

First, we determine the reference tokenizer as the tokenizer of the top-1 LLM (Equation 5). Then, we follow the Minimum Edit Distance (MinED) approach (Fu et al., 2023; Wan et al., 2024), where each token of the reference tokenizer is mapped to its textually closest token of another tokenizer, e.g., "get" to "gets". We tokenize the sequence with the reference tokenizer and use the MinED token mappings to obtain a valid input for another LLM. With the new input, we obtain the output logits of the LLM over its vocabulary. Then, we map the output logits back to the reference tokenizer's vocabulary via MinED in order to

perform logit fusion via Equation 1. MinED mappings between tokenizers are precomputed before inference. We note that aligning different tokenizers is an open problem in NLP, and more effective alignment techniques may further improve PackLLM.

## 4 Language Modeling Experiments

The first research question that we answer is:
**RQ1**. *Is perplexity a good measure for determining the LLM importance weights at test-time?*

We compare with expertise-based model ensemble approaches cBTM(Gururangan et al., 2023) and DExperts (Liu et al., 2021), which, unlike PackLLM, do not rely on perplexity.

**cBTM** performs model fusion at the output probabilities, where the weight of each expert is determined based on the tf-idf similarity between the given prompt and the expert's pretraining data. In particular, cBTM computes weights $\{\lambda_1, \ldots, \lambda_K\}$ for a test prompt $P$ for test-time fusion as

$$\lambda_k \propto \text{softmax}\big( - d(\boldsymbol{h}_P, \boldsymbol{h}_k)\big) \text{ and } p(X_t|x_{<t}) = \sum_k \lambda_k p_k(X_t|x_{<t}), \tag{8}$$

where $d(\cdot, \cdot)$ is a Euclidean distance, $\boldsymbol{h}_P$ is the tf-idf embedding of the prompt, and $\boldsymbol{h}_k$ is a representation of the pretraining data of model $k$. Note that Equation 8 does not use perplexity, but relies on data embedding similarity.

**DExperts** and its follow-up works (Liu et al., 2024; Li et al., 2024b) employ an additional base model $M$ and an anti-expert model $M^-$ to fuse knowledge from an expert model. In particular, the fusion is performed as

$$\boldsymbol{p} = \text{softmax}\big(\boldsymbol{s}_M + \lambda[\boldsymbol{s}_{M_1^*} - \boldsymbol{s}_{M^-}]\big), \tag{9}$$

where $M_1^*$ is the top-1 expert from the seed LLMs, and $\lambda$ is a hyperparameter (default value is $\lambda = 1$). DExperts uses fixed LLM weights.

### 4.1 Experimental Setup

We follow the cBTM experimental setting, where C4 (Roberts et al., 2019) and S2ORC (Lo et al., 2019) datasets are clustered into $K \in \{1, 2, 4, 8, 16\}$ clusters, and $K$ OPT-1.3B models (Zhang et al., 2022) are trained on each cluster separately. Therefore, each OPT model is an expert on a specific semantic cluster of the training documents. The clusters are computed based on the tf-idf embeddings of the train data and the embedding of the cluster centroid is used as $\boldsymbol{h}_k$ in Equation 8 of cBTM. Overall, we experiment with a total of 65 LLMs. To compare with DExperts, we employ the OPT-6.7B, OPT-13B and OPT-30B models as additional base models $M$. Following DExperts, the anti-expert is the original OPT-1.3B model. For each dataset, we report language modeling perplexity on 200 randomly-sampled held out documents. For each test document, the prompt $P$ consists of the 32 first tokens or the first 20% of the sequence's tokens, whichever is shorter; perplexity is evaluated on the rest of tokens.

### 4.2 Results

Figure 3a shows perplexity results when we compare PackLLM with the weighted ensemble approach of cBTM. PackLLM outperforms cBTM under all values of the number of expert LLMs $K$. The performance differences grow as the number of LLMs increases and the largest improvement is observed when we have $K = 16$ experts. PackLLM takes into account the perplexity of the underlying experts to estimate their expertise, while cBTM relies on off-the-shelf similarity embeddings and may not successfully capture the LLMs' knowledge. PackLLM$_{\text{sim}}$ and PackLLM$_{\text{opt}}$ perform similarly on the C4 dataset, while PackLLM$_{\text{sim}}$ scales better on S2ORC dataset, which has shorter prompts. The results show that our perplexity-guided fusion is effective and scales better as we increase the number of expert models.

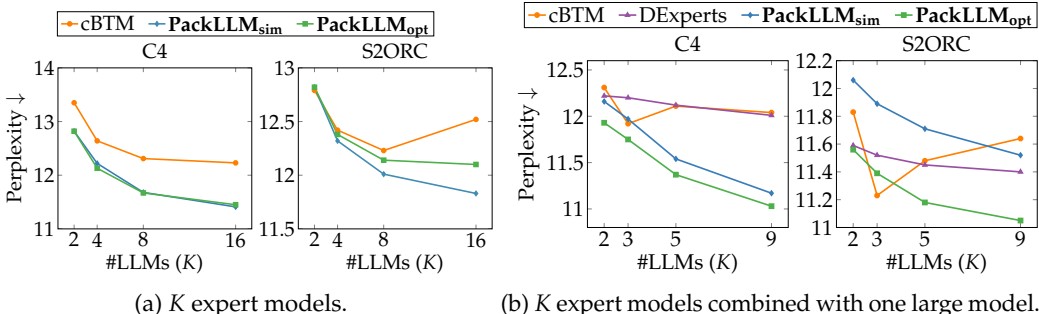

Figure 3: Perplexity result comparison (the lower, the better) on C4 and S2ORC datasets.

Figure 3b shows averaged perplexity results when we add additional large models (OPT-6.7B OPT-13B, OPT-30B) to the set of seed expert LLMs (C4/S2ORC-tuned OPT1.3B models), evaluating whether the fusion approaches can effectively incorporate larger general-purpose LLMs. PackLLM outperforms cBTM and DExperts, and the improvement becomes larger as we increase the number of expert LLMs (value $K$). Moreover, PackLLM$_{opt}$ outperforms PackLLM$_{sim}$ in both C4 and S2ORC; its benefit lies on its perplexity optimization, which effectively combines knowledge from expert models with larger models to boost performance during inference. The result show that PackLLM is effective with LLMs of varying sizes and expertise.

## 5 Downstream Tasks Experiments

The following research questions that we answer are:
**RQ2**. *How does PackLLM compare against other test-time fusion approaches in downstream tasks?*
**RQ3**. *What is the benefit of PackLLM over learning-based fusion approaches?*

### 5.1 Experimental Setup

**Datasets**. We experiment with a diverse set of tasks. We employ the **MMLU-STEM** (Hendrycks et al., 2021) dataset to evaluate performance in knowledge intensive tasks, such college STEM exams. We employ ARC (Clark et al., 2018), BoolQ (Clark et al., 2019), HellaSwag (Zellers et al., 2019), and OpenBookQA (Mihaylov et al., 2018) datasets to evaluate performance on **Commonsense** reasoning. Furthermore, we employ MMLU-Health, PubMedQA (Jin et al., 2019), USMLE (Jin et al., 2021) datasets to evaluate performance on domain-specific knowledge (**Medicine**). In addition, we evaluate on miscellaneous (**Misc.**) classification tasks (Mavromatis et al., 2023), such as topic classification, hatespeech detection, and sentiment analysis. Further details can be found in Appendix C. We report accuracy on question answering and classification as the evaluation metric.

**Seed LLMs**. For MMLU-STEM and Commonsense tasks, we employ widely used LLMs, such as Mistral-7B (Jiang et al., 2023a) and LLaMa2-7B (Touvron et al., 2023). For Medicine tasks, we employ specialized LLMs for biomedical data, such as AdaptLLM (Cheng et al., 2023) and BioMistral (Labrak et al., 2024). For Misc. tasks, we use models with complementary expertise (Gururangan et al., 2023). Overall, we experiment with a total of 52 LLMs (see Appendix D).

**Baselines**. We compare with the following test-time and learning-based fusion approaches. (i) **Top1-PPL** selects the most relevant LLM for a test prompt $P$. We use prompt perplexity (Equation 5) to determine the top-1 LLM. (ii) **Ensemble** performs uniform aggregation of the LLM's logits where every weight $\lambda_k$ has value $\lambda_k = \frac{1}{K}$. (iii) **FuseLLM** (Wan et al., 2024) uses the output logits of the seed LLMs to train a fusion LLM via knowledge distillation. However, it requires an external corpus for causal language modeling pretraining and cannot encompass new LLMs at test-time. (iv) **FoE** (Wang et al., 2023) trains a neural network classifier to select the most relevant expert from the seed LLMs for a test prompt.

Table 1: Main Results. PackLLM outperforms other test-time fusion approaches in 25 tasks and with different seed LLMs.

| Test-Time Fusion | MMLU STEM 5 tasks (5-shot) | | | Commonsense 4 tasks (0-shot) | | | Medicine 10 tasks (0/5-shot) | | Misc. 6 tasks (5-shot) | Average |
|---|---|---|---|---|---|---|---|---|---|---|
| | #Seed LLMs K (#Total Parameters) | | | | | | | | | |
| | 2 (14B) | 4 (23.7B) | 6 (34.7B) | 2 (14B) | 4 (23.7B) | 6 (34.7B) | 3 (27B) | 4 (28B) | 12 (15.6B) | |
| cBTM | – | – | – | – | – | – | – | – | 61.95 | – |
| DExperts | – | – | – | – | – | – | 44.37 | – | – | – |
| Ensemble | 41.29 | 43.54 | 43.51 | 66.30 | 64.45 | 62.10 | 42.22 | 51.59 | 62.53 | 53.05 |
| Top1-PPL | 37.59 | 40.73 | 44.67 | 61.32 | 64.45 | 65.31 | 46.83 | 50.11 | 63.95 | 52.77 |
| **PackLLM$_{sim}$** | 42.16 | 44.22 | 44.48 | 63.84 | 69.23 | 68.50 | 43.55 | 53.73 | 63.20 | 54.77 |
| **PackLLM$_{opt}$** | 42.30 | 45.02 | 45.60 | 61.81 | 67.67 | 67.58 | 47.21 | 53.88 | 63.38 | **54.94** |

– : cBTM and DExperts cannot handle arbitrary seed LLMs.

Table 2: Results on LLM fusion approaches. PackLLM leverages recently released LLMs and outperforms learning-based approaches.

| Seed LLMs → (K, #Params) release date | FuseLLM LLMs (3, 21B) before 07/2023 | FoE LLMs (15, 93B) before 08/2023 | Recent LLMs (4, 23.7B) 07-12/2023 | Recent SLMs* (3, 6.5B) 12/2023-02/2024 |
|---|---|---|---|---|
| | STEM / Com. | STEM / Com. | STEM / Com. | STEM / Com. |
| *Learning-based fusion* | | | | |
| FuseLLM-7B (Wan et al., 2024) | 33.08 / 58.59 | (↻) / (↻) | (↻) / (↻) | (↻) / (↻) |
| FoE (Wang et al., 2023) | (↻) / (↻, †) | 40.65 / (†) | (↻) / (↻, †) | (↻) / (↻, †) |
| *Test-time fusion* | | | | |
| **PackLLM** | 31.21 / 54.88 | 40.26 / 67.05 | 45.02 / 69.23 | 37.00 / 68.25 |

∗ : We use the term SLMs (Small Language Models) to emphasize that their size is below 3B.
↻ : These approaches require retraining with the new seed LLMs before test-time fusion.
† : These approaches require additional annotated data for training.

However, it requires annotated data, e.g., validation data for the downstream tasks, which limits its applicability to new tasks and LLMs. We also compare with (v) **cBTM** and (vi) **DExperts** when possible, as they cannot handle arbitrary seed LLMs (see Appendix B).

## 5.2 Results

Table 1 compares PackLLM with competing test-time approaches, such as top-1 expert selection based on perplexity (Top1-PPL) and uniform ensemble (Ensemble). PackLLM outperforms these approaches by 1.72–1.89% accuracy points, averaged over different seed LLMs in 25 tasks. The improvement becomes larger when using a greater number of LLMs, e.g., in commonsense (+3.19% accuracy points) and medicine (+2.29% accuracy points) tasks. Uniform ensemble may be negatively impacted when we increase the number of LLMs (not all LLMs are equally strong), while Top1-PPL improves with more LLMs. PackLLM$_{opt}$ outperforms PackLLM$_{sim}$ in all tasks, except for Commonsense, where the input prompts are shorter (0-shot input). Moreover, PackLLM outperforms cBTM and DExperts in the cases they can compare against. The results show that PackLLM is an effective fusion approach, outperforming competing test-time approaches by up to 1.89% accuracy points, on average.

In Table 2, we compare PackLLM with learning-based fusion approaches. FuseLLM trains a fusion LLM that learns to combine knowledge from the seed LLMs over diverse pretraining data, while FoE trains a classifier to select the best expert LLM based on validation data. As a result, both FuseLLM and FoE have an advantage over our learning-free method when using the same seed LLMs, leading to 0.39–3.71% more accuracy points. However, PackLLM does not require any training or validation data and can quickly adapt as new LLMs are released. For example, PackLLM improves over FuseLLM and FoE by 4.37–11.94% accuracy points when using four recently released 7B LLMs. In addition, by using strong small LMs (Li et al., 2023), PackLLM outperforms FuseLLM by 3.92–10.66% accuracy points without incurring additional computational costs (equal number of total LLM parameters). Full results of this section can be found in Appendix E.

Table 3: PackLLM comparison with (A) selecting the best model, (B) selecting the largest model, and (C) model merging (average weighting). Due to space limitations, details on experimental setup are provided in Appendix F.

| (A) | | STEM / Com. (acc) *Different Vocab* | STEM / Com. (acc) *Same Vocab* | C4 / S2ORC (ppl) *Same Vocab* |
|---|---|---|---|---|
| | Best LLM | **33.0 / 56.3** | 34.8 / 59.1 | 12.53 / 11.46 |
| | PackLLM | 31.2 / 54.9 | **40.3 / 67.1** | **11.03 / 11.05** |
| (B) | | Medicine (acc) | STEM (acc) | C4 (ppl) |
| | Largest LLM | 49.9 | 33.1 | 12.71 |
| | PackLLM | **52.7** | **36.3** | **11.13** |
| (C) | | Com. (acc) | C4 (ppl) | S2ORC (ppl) |
| | Model Merging | 55.2 | 13.20 | 12.92 |
| | PackLLM | **57.1** | **12.12** | **12.32** |

## 6 Comparison with Model Selection and Merging

In this section, we compare PackLLM with respect to (A) selecting the best model, (B) selecting the largest model, and (C) model merging. The results are summarized in Table 3.

**PackLLM vs. Best Model**. We experiment with the best LLM (as evaluated on test data) on STEM and Commonsense tasks. As Table 3A shows, the best LLM works better when having few LLMs with different vocabularies, but when we have a large number of LLMs with the same vocabulary, PackLLM outperforms the best LLM. By developing better tokenization mapping methods (Section 3.4), PackLLM may become more effective in the first scenario. To further verify our conclusions, we experiment on the C4 and S2ORC language modeling tasks, where we have a large number of expert OPT models sharing the same vocabulary. In this case, PackLLM also outperforms the Best LLM (selected as the top1 expert).

**PackLLM vs. Largest Model**. Selecting a larger LLM may yield better generalization than selecting smaller models with lower perplexity. However, smaller LMs are easy to fine-tune on in-domain data which can provide complementary knowledge (e.g., medicine data). In such cases, model fusion yields better results. To demonstrate this, we compare PackLLM with selecting the largest model from the set of models on three different tasks; Medicine, MMLU-STEM, and language modeling on C4 in Table 3B. The results show that PackLLM yields better performance than selecting the largest model in all settings. We provide example cases, where PackLLM improves a large model in Appendix G.

**PackLLM vs. Model Merging**. Although it is not always possible to merge models as a fusion approach, model merging has lower inference costs compared to ensemble approaches. We provide results by comparing PackLLM with model merging (average weighting) on commonsense reasoning and language modeling tasks in Table 3C. The results that PackLLM outperforms model merging across these two tasks.

## 7 Ablation Studies

In this section, we ablate PackLLM$_{\text{sim}}$ and PackLLM$_{\text{opt}}$ with respect to their derived LLM importance weights, their sensitivity on the input prompt, and their time cost.

**Importance Weights**. Figure 4a shows the derived importance weights $\lambda_k$ by the two approaches for four different tasks. PackLLM$_{\text{sim}}$ calculates the weights based on perplexity scores (Equation 4) and thus, fluctuations in perplexity scores among datasets and LLMs result in fluctuations in the induced weights. For example, the top-1 weights range within $[0.54, 0.89]$ and top-2 weights range within $[0.08, 0.37]$. On the other hand, PackLLM$_{\text{opt}}$ solves an optimization of minimizing perplexity to determine weights $\lambda_k$, which results in weights of smaller range. For example, the top-1 weights range within $[0.56, 0.76]$ and top-2 weights range within $[0.16, 0.31]$.

**Prompt Length**. Figure 4b compares cBTM, PackLLM$_{\text{sim}}$, and PackLLM$_{\text{opt}}$ with respect to the length of the input prompt. cBTM does not take into account how well the LLMs

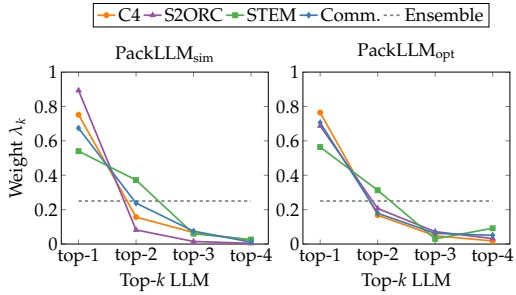

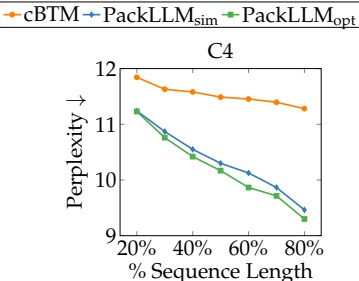

(a) Importance weight values for PackLLM$_{\text{sim}}$ (left) and PackLLM$_{\text{opt}}$ (right) in 4 tasks.

(b) Perplexity results with respect to input prompt length (x-axis).

Figure 4: Ablation studies of PackLLM$_{\text{sim}}$ and PackLLM$_{\text{opt}}$.

understand the input and shows minor improvements with longer prompts. On the other hand, both PackLLM$_{\text{sim}}$ and PackLLM$_{\text{opt}}$ significantly improve as we increase the length of the input. Moreover, PackLLM$_{\text{opt}}$ outperforms PackLLM$_{\text{sim}}$ due to its optimization framework, which specializes into understanding the input better.

**Computation Time**. Table 4 compares the time cost of different optimization approaches for determining the weights $\lambda_k$ on the C4 dataset. PackLLM$_{\text{sim}}$ does not solve an optimization problem and thus does not result in any time overhead (row "None" in Table 4). However, it is not ensured that it can achieve the best performance (PPL). On the other hand, PackLLM$_{\text{opt}}$ employs a greedy solution to its optimization problem

Table 4: Weight Optimization Time Cost.

| PPL Opt. #LLMs | Opt. Time ↓ (sec.) 3 / 5 | Perplexity ↓ 3 / 5 |
|---|---|---|
| **Grid Search** | | |
| None | 0 / 0 | 11.96 / 11.54 |
| Greedy | 2 / 4 | 11.77 / 11.40 |
| Exhaustive | 14 / 902 | 11.76 / 11.38 |

(greedy grid search), improving downstream performance (perplexity), while maintaining a linear time complexity with respect to the number of seed LLMs (#LLMs), i.e., 2-4 secs when using 2-5 LLMs. Although exhaustive grid search can slightly improve performance compared to greedy grid search, its time cost can be prohibitive. For example, when using 5 LLMs, exhaustive grid search results in a explosive time cost of 902 secs.

Furthermore, we compare the inference time cost of PackLLM with the simple ensemble approach. PackLLM computes the output logits by each LLM once (as in the Ensemble baseline), which are then used for determining the importance weights as shown in Algorithm 1 of Appendix A (no additional LLM forward passes are required). We provide the time measurements for $K = 5$ LLMs per input on a A100 GPU in Table 5. As evaluated, PackLLM does not incur significant time costs.

Table 5: Inference Time Cost.

| | Time (secs) |
|---|---|
| Single forward pass | 0.06 |
| Ensemble ($K = 5$ forward passes) | 0.30 |
| PackLLM ($K = 5$ forward passes) | 0.32 |

PackLLM's limitations are discussed in Appendix H.

# 8 Conclusion

We introduce PackLLM, an effective method for test-time fusion that leverages each LLM's expertise, given an input prompt. PackLLM performs model fusion by solving an optimization problem for determining each LLM's importance, so that perplexity over the input prompt is minimized. We conduct experiments with over 100 total LLMs on a diverse set of tasks. Experimental results show that (i) perplexity is a reliable measure for LLM fusion, (ii) PackLLM outperforms test-time fusion baselines by 1.89% accuracy points, (iii) PackLLM can leverage new LLMs to improve performance over learning-based fusion approaches by 3.92–11.94% accuracy points, and (iv) PackLLM benefits over selecting the best or largest model and model merging in certain cases.

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

# Appendix

## A  Greedy Algorithm

We present the greedy algorithm for prompt perplexity minimization in Algorithm 1.

---

**Algorithm 1** Greedy Perplexity Minimization for Test-Time Fusion.

---

1: **Input:** Test Prompt $P = (x_0, x_1, \ldots, x_t)$, Seed LLMs $\{M_1, \ldots, M_K\}$.      *{test-time}*

2: **for all** $k \in K$ **do**
3:      Compute $\text{PPL}_k(P)$.      *{perplexity estimation}*
4: **end for**
5: Sort LLMs via: $[M_1^*, \ldots, M_K^*] = argsort(\text{PPL}_1(P), \ldots, \text{PPL}_K(P))$.
6: Set $\lambda_1^* = 1$ and $\lambda_{k>1}^* = 0$.      *{weight initialization}*

7: Compute logits $\boldsymbol{s}^{(1)}$ using $M_1^*$.      *{forward pass}*
8: **for all** $k \in [1, K-1]$ **do**
9:      Compute logits $\boldsymbol{s}^{(k+1)}$ using $M_k^*$.      *{forward pass}*
10:      **for** $\lambda \in [0, 1, step = 0.05]$ **do**
11:          Compute $\mathcal{L}_\lambda = -\frac{1}{t}\sum_i^t \log \text{softmax}\left(\lambda \boldsymbol{s}^{(k)}(x_i | x_{<i}) + (1-\lambda)\boldsymbol{s}^{(k+1)}(x_i | x_{<i})\right)$.    *{grid search}*
12:      **end for**
13:      Select $\lambda^{(k)} = \arg\min_\lambda \mathcal{L}_\lambda$.      *{loss minimization}*
14:      Set $\boldsymbol{s}^{(k)} = \lambda^{(k)}\boldsymbol{s}^{(k)} + (1-\lambda^{(k)})\boldsymbol{s}^{(k+1)}$.      *{greedy fusion}*
15:      Set $\lambda_{<k+1}^* = \lambda^{(k)} \cdot \lambda_{<k+1}^*$ and $\lambda_{k+1}^* = (1-\lambda^{(k)})$.      *{weight update}*
16: **end for**

17: **Output:** Weights $\{\lambda_1^*, \ldots, \lambda_K^*\}$ for test input $P$.      *{weights for $\{M_1^*, \ldots, M_K^*\}$}*

18: **Generation:** Sample with token probability $p(x_{t'} | x_{<t'}) = \text{softmax}\left(s(x_{t'} | x_{<t'})\right)$, where $s = \lambda_1^* \boldsymbol{s}_{M_1^*} + \cdots + \lambda_K^* \boldsymbol{s}_{M_K^*}$.      *{LLM fusion}*

---

## B  PackLLM vs. cBTM & DExperts

We present some challenges on comparing with cBTM and DExperts:

- cBTM controls the importance weights based on the distance from the input prompt to the pretraining data of each (expert) LLM. In order to produce meaningful weights cBTM assumes each expert was trained on a different domain and that the pretraining data for all experts is known. These necessary assumptions limit the applicability of this approach to arbitrary LLMs where the pretraining data is unknown (e.g. LLama) or where there is likely significant overlap in pretraining domain (e.g. Llama + Mistral).

  or example, if we use cBTM's LLMs, they perform  25% at accuracy on MMLU, which is close to random answer selection. Generalizing cBTM to other model families for producing full results in Table 1 requires (i) proper data selection, and (ii) further model fine-tuning, which is limiting regarding compute resources.

- DExperts combines three LLMs (see Equation 9): (i) a large model $M$, (ii) a fine-tuned (in-domain expert) $M_1$ and an untuned/anti-expert (out of domain) $M^-$. This approach does not extend to evaluating model fusion with a variable number of seed LLMs.

If all experts are known, PackLLM outperforms cBTM (Misc in Table 1) and DExperts (Medicine in Table 1).

Following, we highlight the key difference between cBTM, DExperts, and PackLLM (ours) regarding how they compute the importance weights:

- cBTM: Importance weights are **data-dependent** (Equation 8; using tf-idf embeddings); the weights would be similar for LLaMA-7B and LLaMA-70B as the two models have similar pretraining data.
- DExperts: Importance weights are **fixed** (Equation 9).
- PackLLM: Importance weights are **model-dependent** (perplexity-based), capturing how well each model understands the prompt; LLaMA-70B would be assigned a higher weight than LLaMA-7B.

Figure 3 verifies the benefit of model-dependent weights (PackLLM vs. cBTM & DExperts).

## C   Dataset Details

All datasets can be found in HuggingFace Hub.

C4 (Roberts et al., 2019) is s a publicly available distribution of a Common Crawl snapshot. Following cBTM, we use 168B tokens of the no blocklist version (en.noblocklist) that is out of distribution to the OPT model's pretraining corpus. S2ORC (Lo et al., 2019) is a publicly available corpus of full-text academic papers from the Semantic Scholar. The corpus spans 20 fields of study (e.g., Biology, Computer Science, Art). Following cBTM, the training data consists of 168B tokens over multiple epochs. For each dataset, we report language modeling perplexity on 200 randomly-sampled held out documents. For each test document, the prompt P consists of the 32 first tokens or the first 20% of the sequence's tokens, whichever is shorter; perplexity is evaluated on the rest of tokens.

MMLU (Hendrycks et al., 2021) consists of knowledge intensive tasks, such as STEM exams, that have a question and four multiple choice questions. The prompt template is "`Question: {QUESTION} Choices: {Choices} Answer: `". We use the default 5-shot examples and evaluate on the test data.

We employ ARC (Clark et al., 2018), BoolQ (Clark et al., 2019), HellaSwag (Zellers et al., 2019), and OpenBookQA (Mihaylov et al., 2018) datasets to evaluate performance on commonsense reasoning. We use 0-shot prompts with the template "`Question: {QUESTION} Choices: {CHOICES} Answer: `" for ARC, BoolQ and OpenBookQA, and continuation "`The topic is about {TOPC}. {CTX-A} {CTX-B}`" for HellaSwag. We randomly sample 256 test data for evaluation. BoolQA is a binary classification, while ARC, HellaSwag, and OpenBookQA usually have four answer choices.

For PubMedQA (Jin et al., 2019) and USMLE (Jin et al., 2021), we follow (Cheng et al., 2023). The template is "`Question: {QUESTION} Choices: {CHOICES} Answer: `" for PubMedQA and "`{CONTEXT} {QUESTION} {ANSWER}`" for USMLE. These datasets have four-way multiple choice answers. We randomly sample 256 test data for evaluation and use 0-shot prompts.

Regarding miscellaneous classification tasks, we experiment with AGNews (Zhang et al., 2015) (topic classification), SST2 (Socher et al., 2013) (sentiment analysis), Amazon (McAuley & Leskovec, 2013) (sentiment analysis), and SemEval (Basile et al., 2019) (Twitter hate-speech/sentiment). We follow (Mavromatis et al., 2023) which constructs a pool of 30 annotated data, and the 5 closest examples of the pool (via embedding similarity) are used as few shot examples for a test input. We evaluate on 256 randomly sampled test data.

## D   Seed LLMs Details

In this section, we provide the full list of the LLMs we use.

**MMLU STEM**. We use the following LLMs:

- 2 Seed LLMs:
- `meta-llama/Llama-2-7b-hf`

- `mistralai/Mistral-7B-v0.1`

- 4 Seed LLMs:
- `meta-llama/Llama-2-7b-hf`
- `mistralai/Mistral-7B-v0.1`
- `microsoft/phi-2`
- `Deci/DeciLM-7B`

- 6 Seed LLMs:
- `meta-llama/Llama-2-7b-hf`
- `mistralai/Mistral-7B-v0.1`
- `microsoft/phi-2`
- `Deci/DeciLM-7B`
- `BioMistral/BioMistral-7B`
- `EleutherAI/llemma-7b`

**Commonsense**. We use the following LLMs:

- 2 Seed LLMs: Same as MMLU-STEM.
- 4 Seed LLMs: Same as MMLU-STEM.
- 6 Seed LLMs:
- `meta-llama/Llama-2-7b-hf`
- `mistralai/Mistral-7B-v0.1`
- `microsoft/phi-2`
- `Deci/DeciLM-7B`
- `Wanfq/FuseLLM-7B`
- `openlm-research/open-llama-7b-v2`

**Medicine**. We use the following LLMs:

- 3 Seed LLMs:
- `AdaptLLM/medicine-LLM` (expert)
- `luodian/llama-7b-hf` (anti-expert)
- `meta-llama/Llama-2-13b-hf` (base)

- 4 Seed LLMs:
- `BioMistral/BioMistral-7B`
- `llSourcell/medllama2-7b`
- `AdaptLLM/medicine-LLM`
- `chaoyi-wu/PMC-LLAMA-7B`

**Miscellaneous**. We use 12 expert OPT-1.3B models on trained on the C4 dataset by (Guru-rangan et al., 2023). We combine together 8 cluster experts and 4 cluster experts.

**FuseLLM LLMs**. FuseLLM uses the following 3 LLMs:

- `openlm-research/open-llama-7b-v2`
- `mosaicml/mpt-7b`
- `meta-llama/Llama-2-7b-hf`

**FoE LLMs**. FuseLLM uses the following 15 LLMs:

- `Aspik101/trurl-2-7b-pl-instruct_unload`
- `Charlie911/vicuna-7b-v1.5-lora-mctaco`
- `Fredithefish/RedPajama-INCITE-Chat-3B-Instruction-Tuning-with-GPT-4`
- `GOAT-AI/GOAT-7B-Community`
- `TheTravellingEngineer/bloom-1b1-RLHF`
- `ashercn97/manatee-7b`
- `garage-bAInd/Platypus2-7B`
- `golaxy/gogpt-7b-bloom`
- `julianweng/Llama-2-7b-chat-orcah`
- `lmsys/vicuna-7b-v1.3`
- `lmsys/vicuna-7b-v1.5-16k`
- `medalpaca/medalpaca-7b`
- `rombodawg/LosslessMegaCoder-llama2-7b-mini`
- `togethercomputer/GPT-JT-6B-v0`
- `togethercomputer/GPT-JT-6B-v1`

**Recent LLMs**. As recently released LLMs, we use the following:

- `meta-llama/Llama-2-7b-hf`
- `mistralai/Mistral-7B-v0.1`
- `microsoft/phi-2`
- `Deci/DeciLM-7B`

**Recent SLMs**. As the recently released SLMs, we use the following:

- `google/gemma-2b`
- `microsoft/phi-2`
- `Qwen/Qwen1.5-1.8B`

## E  Full Results

Table 6 provides full results on MMLU-STEM tasks.

Table 7 provides full results on commonsense tasks.

Table 8 provided full results on medicine tasks. Further MMLU-Health results are provided in Table 9.

Table 10 provides full results on miscellaneous classification tasks.

## F  Setup of Comparison with Model Selection and Merging

We provide the experimental setup of the results in Table 3:

- **(A) STEM / Com. (different vocab)**:
  - 5 MMLU-STEM tasks and 4 Commonsense tasks.
  - 3 seed LLMs of different vocab (Fuse LLMs).
  - Best LLM: LLaMA2-7B.
- **(A) STEM / Com. (same vocab)**:

Table 6: Full results on MMLU-STEM (College) tasks.

| | Biology | Chemistry | CS | Math | Physics | Average |
|---|---|---|---|---|---|---|
| | 2 Seed LLMs (14B) | | | | | |
| Ensemble | 59.02 | 42.00 | 45.00 | 32.00 | 28.43 | 41.29 |
| Top1-PPL | 46.52 | 40.00 | 37.00 | 33.00 | 31.37 | 37.59 |
| **PackLLMsim** | 53.47 | 45.00 | 45.00 | 35.00 | 32.35 | 42.16 |
| **PackLLMopt** | 54.17 | 46.00 | 45.00 | 33.00 | 33.33 | **42.30** |
| | 4 Seed LLMs (23.7B) | | | | | |
| Ensemble | 60.41 | 49.00 | 41.00 | 32.00 | 35.29 | 43.54 |
| Top1-PPL | 58.33 | 34.00 | 38.00 | 39.00 | 34.31 | 40.73 |
| **PackLLMsim** | 61.80 | 41.00 | 47.00 | 37.00 | 34.31 | 44.22 |
| **PackLLMopt** | 63.88 | 40.00 | 41.00 | 41.00 | 39.21 | **45.02** |
| | 6 Seed LLMs (34.7B) | | | | | |
| Ensemble | 58.33 | 43.00 | 44.00 | 34.00 | 38.23 | 43.51 |
| Top1-PPL | 58.33 | 39.00 | 39.00 | 37.00 | 50.00 | 44.67 |
| **PackLLMsim** | 63.19 | 42.00 | 44.00 | 35.00 | 38.23 | 44.48 |
| **PackLLMopt** | 63.88 | 44.00 | 46.00 | 31.00 | 43.13 | **45.60** |
| | 3 Seed LLMs, same with FuseLLM (21B) | | | | | |
| Ensemble | 40.97 | 33.00 | 34.00 | 26.00 | 23.52 | **31.49** |
| Top1-PPL | 43.05 | 27.00 | 31.00 | 30.00 | 20.58 | 30.32 |
| **PackLLMsim** | 43.75 | 29.00 | 32.00 | 30.00 | 21.56 | 31.26 |
| **PackLLMopt** | 44.44 | 31.00 | 32.00 | 31.00 | 17.64 | 31.21 |
| | 15 Seed LLMs, same with FoE (93B) | | | | | |
| Ensemble | 46.06 | 34.75 | 36.77 | 33.40 | 26.47 | 35.49 |
| Top1-PPL | 47.91 | 39.00 | 36.00 | 30.00 | 24.50 | 35.48 |
| **PackLLMsim** | 53.47 | 37.00 | 37.00 | 41.00 | 25.49 | 38.79 |
| **PackLLMopt** | 54.86 | 38.00 | 43.00 | 39.00 | 26.47 | **40.26** |
| | 3 Seed SLMs (6.5B) | | | | | |
| Ensemble | 49.30 | 34.00 | 38.00 | 31.00 | 21.56 | 34.77 |
| Top1-PPL | 56.25 | 37.00 | 35.00 | 32.00 | 24.50 | 36.95 |
| **PackLLMsim** | 55.55 | 35.00 | 37.00 | 31.00 | 26.47 | **37.00** |
| **PackLLMopt** | 56.94 | 30.00 | 40.00 | 29.00 | 27.45 | 36.68 |
| FuseLLM (7B) | 40.97 | 25.00 | 39.00 | 33.00 | 27.45 | 33.08 |
| FoE (93B) | 54.55 | 36.00 | 40.00 | 42.00 | 30.69 | 40.65 |

Table 7: Full results on Commonsense tasks.

| | ARC-Challenge | BoolQ | HellaSwag | OpenBookQA | Average |
|---|---|---|---|---|---|
| | 2 Seed LLMs (14B) | | | | |
| Ensemble | 54.29 | 80.46 | 74.60 | 55.85 | **66.30** |
| Top1-PPL | 41.01 | 78.51 | 74.60 | 51.17 | 61.32 |
| **PackLLMsim** | 44.43 | 81.25 | 75.39 | 54.29 | 63.84 |
| **PackLLMopt** | 42.96 | 78.90 | 75.00 | 50.39 | 61.81 |
| | 4 Seed LLMs (23.7B) | | | | |
| Ensemble | 52.34 | 76.56 | 76.17 | 52.73 | 64.45 |
| Top-1 | 49.60 | 75.00 | 76.56 | 56.64 | 64.45 |
| **PackLLMsim** | 55.85 | 81.25 | 78.90 | 60.93 | **69.23** |
| **PackLLMopt** | 51.95 | 78.51 | 77.73 | 62.50 | 67.67 |
| | 6 Seed LLMs (34.7B) | | | | |
| Ensemble | 46.48 | 76.17 | 74.21 | 51.56 | 62.10 |
| Top1-PPL | 48.04 | 80.01 | 76.56 | 56.64 | 65.31 |
| **PackLLMsim** | 53.31 | 83.20 | 78.90 | 58.59 | **68.50** |
| **PackLLMopt** | 50.78 | 79.29 | 76.56 | 63.67 | 67.58 |
| | 3 Seed LLMs, same as FuseLLM (21B) | | | | |
| Ensemble | 30.85 | 76.95 | 70.31 | 38.67 | 54.19 |
| Top1-PPL | 29.29 | 75.00 | 74.21 | 38.67 | 53.60 |
| **PackLLMsim** | 28.13 | 76.95 | 74.21 | 40.23 | **54.88** |
| **PackLLMopt** | 29.29 | 75.39 | 74.60 | 40.23 | 54.87 |
| | 15 Seed LLMs, same as FoE (93B) | | | | |
| Ensemble | 47.90 | 73.39 | 72.60 | 52.06 | 61.48 |
| Top1-PPL | 33.59 | 72.65 | 74.21 | 47.26 | 56.92 |
| **PackLLMsim** | 54.68 | 76.17 | 74.21 | 63.17 | **67.05** |
| **PackLLMopt** | 48.82 | 69.53 | 75.39 | 55.07 | 62.20 |
| | 3 Seed SLMs (6.5B) | | | | |
| Ensemble | 48.88 | 66.79 | 60.93 | 46.87 | 55.86 |
| Top1-PPL | 63.67 | 77.34 | 67.96 | 61.32 | 67.57 |
| **PackLLMsim** | 61.71 | 77.34 | 69.53 | 58.98 | 66.89 |
| **PackLLMopt** | 62.50 | 79.68 | 69.14 | 61.71 | **68.25** |
| FuseLLM (7B) | 36.32 | 77.73 | 75.78 | 44.53 | 58.59 |

Table 8: Full results on Medicine tasks.

| | MMLU-Health
*8 tasks, 5-shot* | PubMedQA
*0-shot* | USMLE
*0-shot* | **Average**
(weighted) |
|---|---|---|---|---|
| | 3 Seed LLMs (with anti-expert) | | | |
| DExperts | 44.17 | 66.40 | 33.98 | 48.18 (45.37) |
| Ensemble | 40.31 | 68.75 | 31.03 | 46.69 (42.22) |
| Top1-PPL | 45.70 | 67.96 | 34.76 | 49.47 (46.83) |
| **PackLLM$_{sim}$** | 41.85 | 68.35 | 32.42 | 47.54 (43.55) |
| **PackLLM$_{opt}$** | 46.13 | 69.14 | 33.98 | 49.75 (**47.21**) |
| | 4 Seed LLMs | | | |
| Ensemble | 51.45 | 69.53 | 34.76 | 51.91 (51.59) |
| Top1-PPL | 51.61 | 54.29 | 33.98 | 46.62 (50.11) |
| **PackLLM$_{sim}$** | 53.22 | 67.57 | 33.98 | 51.59 (53.73) |
| **PackLLM$_{opt}$** | 54.65 | 67.96 | 33.59 | 52.06 (**53.88**) |

Table 9: Full results on MMLU-Health tasks.

| | Anatomy | Clinical | CollegeMed | HumanAge | MedGenetics | Nutrition | ProMed | Virology | **Average** |
|---|---|---|---|---|---|---|---|---|---|
| | 4 Seed LLMs (28B) | | | | | | | | |
| Ensemble | 48.88 | 58.49 | 42.77 | 55.15 | 59.00 | 58.49 | 44.85 | 43.97 | 51.45 |
| Top1-PPL | 43.73 | 57.35 | 46.24 | 58.74 | 63.00 | 54.90 | 43.75 | 45.18 | 51.61 |
| **PackLLM$_{sim}$** | 47.40 | 60.00 | 46.24 | 60.98 | 68.00 | 52.94 | 44.48 | 45.78 | 53.22 |
| **PackLLM$_{opt}$** | 50.37 | 62.64 | 45.66 | 61.43 | 67.00 | 57.18 | 47.79 | 45.18 | **54.65** |

Table 10: Results on Miscellaneous classification tasks.

| | AGNews
*Topic* | Ethos
*Hatespeech* | TweetHate
*Hatespeech* | Amazon
*Sentiment* | TweetComplain
*Sentiment* | SST2
*Sentiment* | **Average** |
|---|---|---|---|---|---|---|---|
| | 12 Seed LLMs | | | | | | |
| cBTM | 58.98 | 50.39 | 46.09 | 70.70 | 78.00 | 67.57 | 61.95 |
| Ensemble | 62.10 | 51.17 | 46.48 | 69.92 | 76.00 | 69.53 | 62.53 |
| Top1-PPL | 62.89 | 51.17 | 45.31 | 74.21 | 78.00 | 72.17 | 63.95 |
| **PackLLM$_{sim}$** | 61.71 | 51.56 | 46.48 | 69.92 | 80.00 | 69.53 | 63.20 |
| **PackLLM$_{opt}$** | 60.93 | 55.07 | 46.09 | 70.31 | 78.00 | 69.92 | 63.38 |

- – 5 MMLU-STEM tasks and 4 Commonsense tasks.
- – 15 seed LLMs of same vocab (FoE LLMs).
- – Best LLM: Platypus-7B.
- **(A) C4 and S2ORC**:
  - – 9 OPT models (30B OPT + 8* 1.3 OPT experts) on C4 / S2ORC (cBTM models).
  - – Best LLM: Top1 model selection.
- **(B) Medicine**:
  - – 3 LLMs (LLaMA2-13B, MedLLaMA2-7B, LLaMA2-7B) on 8 Medicine tasks.
  - – Largest LLM: LLaMA2-13B.
- **(B) STEM**:
  - – 4 LLMs (FuseLLM-7B, Gemma-2B, phi2-2.7B, Qwen-1.8B) on 5 MMLU-STEM tasks.
  - – Largest LLM: FuseLLM-7B.
- **(B) C4**:
  - – 5 LLMs (OPT-30B, 4 domain experts OPT-1.3B) on C4 (cBTM models).
  - – Largest LLM: OPT-30B.
- **(C) Com.**:
  - – 2 LLMs (LLaMA2 and FuseLLM 7B models) on 4 Commonsense tasks.
- **(C) C4 / S2ORC**:
  - – 4 LLMs (4 domain experts OPT-1.3B) on C4 / S2ORC (cBTM models).

## G  Example Cases

We include the following two examples where fusing knowledge with PackLLM improves results:

1. • *Input*: "Which of the following contributes to vitamin B12 deficiency in older adults? A. Reduced secretion of intrinsic factor, B. Atrophic gastritis, C. Helicobacter pylori infection, D. All the above"
   • *LLaMA2-13B*: A. Reduced secretion of intrinsic factor (*incorrect → picks the longest answer*)
   • *LLaMA2-13B + PackLLM (MedLLaMA2-7B)*: D. All the above (*correct → identifies all correct answers*)

2. • *Input*: "Contributing your $3,000 to a 401(k) or other "
   • *OPT-6.7B*: "retirement plan today will get you a return of 12-14%" (*hallucination: incorrect return percentage*)
   • *OPT-6.7B + PackLLM (OPT-1.3B tuned model on C4)*: "retirement plan offers some tax advantages, but it's important to take action as early as possible." (*helpful advice*)

## H  Limitations

We summarize two limitations of our approach below.

1. One challenge is discussed in Section 3.4: Equation 1 assumes that the logit vectors can be added together, which requires that the seed LLMs share the same vocabulary. However, this is not always the case, as different LLMs are trained with different vocabularies. While we follow the MinED approach for tokenizer alignment, aligning different tokenizers is an open problem in NLP, and more effective alignment techniques may further improve PackLLM.

2. PackLLM (similar to other fusion approaches) incurs additional inference costs compared to model merging approaches, such as weight averaging. Accelerating the inference time of fusion approaches, such as PackLLM and model ensemble, is a promising future research area.

