# OpenReview forum: "Pack of LLMs: Model Fusion at Test-Time via Perplexity Optimization"
_colmweb.org/COLM/2024/Conference — COLM_

### Official Review · Reviewer_ymUA · 2024-04-18

**Rating:** 7
**Confidence:** 3
**Ethics Flag:** 1

**Summary:**

This paper provides an alternative method for combining the responses from multiple LLMs. Weights were introduced to be used to re-weight the output logits. To determine the criterion for obtaining the optimal weights, the authors first discovered that the perplexity on the given prompts can be already served as a good approach to fuse the outputs of models. Next, the authors propose an greedy optimization-based algorithm that effectively fuses the outputs from different models. Experimental results show that the proposed algorithm can merge the outputs from models better than baseline methods such as ensemble and using only the model with lowest perplexity.

**Reasons To Accept:**

- The motivation is clearly described and the topic is interesting.
- The empirical performance seems to be strong.
- The techniques introduced in this paper are technically sound.

**Reasons To Reject:**

- I think the authors may also want to compare with always picking the responses from the largest model, because selecting the model with the lowest perplexity does not mean that model can generalize well. Selecting always the largest model may yield better performance.
- I would also like to see how this model compared with model merging. I understand it is not always possible to merge models, especially when the number of parameters and the structures are diverse across models. However, I think the authors can at least find several publicly available models fine-tuned from llama-7b on different datasets and see if the proposed algorithm can outperform model merging.

---

> ### Author Rebuttal · Authors · 2024-05-31
>
> Thank you for your review. We address your suggestions below.
>
> **Selecting Largest Model**. Thank you for your suggestion. We agree that selecting a larger LLM may yield better generalization. However, smaller LMs are easy to fine-tune on in-domain data which can provide complementary knowledge (e.g., medicine data). In such cases, model fusion yields better results. To demonstrate this, we compare PackLLM with selecting the largest model from the set of models (and DExperts when applicable) on three different tasks; Medicine, MMLU-STEM, and language modeling. The results show that PackLLM yields better performance than selecting the largest model in all settings. Full results can be found in the tables below.
>
> Setup: 3 LLMs (LLaMA2-13B,  MedLLaMA2-7B, LLaMA2-7B) on 8 Medicine tasks:
> |Fusion| Accuracy|
> |-|-|
> |Select largest (13B) | 49.9|
> |DExperts |50.1|
> |PackLLM |**52.7**|
>
> Setup: 4 LLMs (FuseLLM-7B, Gemma-2B, phi2-2.7B, Qwen-1.8B) on 5 MMLU-STEM tasks.
> |Fusion|Accuracy|
> |-|-|
> |Select largest (7B)|33.1|
> |PackLLM|**36.3**|
>
> Setup: 5 LLMs (OPT-30B,  4 domain experts OPT-1.3B) on C4:
> |Fusion|Perplexity (the lower the better)|
> |-|-|
> |Select largest (30B)|12.71|
> |DExperts|11.67|
> |PackLLM|**11.13**|
>
> **Model Merging**. Thank you for your suggestion. We agree that comparing with model merging can provide new insights; the comparison is also interesting as model merging has lower inference costs compared to fusion approaches. We provide preliminary results by comparing PackLLM with average weight model merging on commonsense reasoning and language modeling tasks. The following tables show results that PackLLM outperforms model merging across these two tasks.
>
> Setup: 2 LLMs (LLaMA2 and FuseLLM 7B models) on 4 Commonsense tasks.
> |Fusion|Accuracy|
> |-|-|
> |Merging (Average Weighting)|55.2|
> |PackLLM|**57.1**|
>
> Setup: 4 LLMs (4 domain experts OPT-1.3B) on C4:
> |Fusion|Perplexity (the lower the better)|
> |-|-|
> |Merging (Average Weighting)|13.20|
> |PackLLM|**12.12**|

---

> > ### Comment · Reviewer_ymUA · 2024-06-05
> >
> > I have read the authors' responses and my concern has been addressed.

---

### Official Review · Reviewer_MKGz · 2024-05-07

**Rating:** 5
**Confidence:** 4
**Ethics Flag:** 1

**Summary:**

This paper presents a test-time LLM fusion method to leverage knowledge from the user-specified LLMs during inference. Specifically, the authors optimize the perplexity over the input prompt to perform model fusion.  The experimental results demonstrate the effectiveness of the proposed method.

**Questions To Authors:**

- Is it possible to complete the results of the test-time baseline experiment?

- It is advised to provide some case studies to show that the proposed ppl-based fusion method is superior to other methods.

**Reasons To Accept:**

This work presents a simple but effective test-time LLM fusing method and provides valuable insights. The paper is well-written and clear. The authors conducted experiments on multiple datasets, illustrating that this method generalizes well to different domains.

**Reasons To Reject:**

In the main experimental results reported in Table 1, the proposed method outperforms the baseline models. However, it is important to note that the experiments involving some test-time fusion methods, such as cBTM and DExperts, are not fully reported. This lack of complete data undermines the reliability of these results. This issue is similar to the one observed in Table 2, where the comparison with learning-based fusion models becomes less convincing due to incomplete experimental data.

---

> ### Author Rebuttal · Authors · 2024-05-31
>
> We thank the reviewer for providing a thorough review and asking insightful questions.
>
> **Comparison with cBTM and DExperts \& Question 1**.
>
> - cBTM controls the importance weights based on the distance from the input prompt to the pretraining data of each (expert) LLM. In order to produce meaningful weights cBTM assumes each expert was trained on a different domain and that the pretraining data for all experts is known. These necessary assumptions limit the applicability of this approach to arbitrary LLMs where the pretraining data is unknown (e.g. LLama) or where there is likely significant overlap in pretraining domain (e.g. Llama + Mistral).
> - DExperts combines three LLMs (see Eq. 9): (i) a large model $M$, (ii) a fine-tuned (in-domain expert) $M_1$ and an untuned/anti-expert (out of domain) $M^-$. This approach does not extend to evaluating model fusion with a variable number of seed LLMs.
>
> Nevertheless, we include comparisons with cBTM and DExperts where applicable e.g.,  in Figure 3 with the setup described in Section 4.1, and Medicine and Misc. in Table 1.
>
> **Comparison with learning-based fusion**.
> As presented in Figure 1, the key motivation of our approach is to combine newer models without requiring retraining. We provide Table 2 to show that modular fusion methods (such as PackLLM) can leverage newly released LLMs (row “release date” in Table 2) to outperform existing learning-based fusion models (e.g. FoE), which, unlike PackLLM, uses additional (pre-)training data and requires retraining.
>
> **Question 2: Example Cases**.
> Thank you for your suggestion. We will include the following two examples where fusing knowledge with PackLLM improves results in the final version:
> - Case A
>
> *Input*: "Which of the following contributes to vitamin B12 deficiency in older adults? A. Reduced secretion of intrinsic factor, B. Atrophic gastritis, C. Helicobacter pylori infection, D. All the above"
>
> *LLaMA2-13B*: A. Reduced secretion of intrinsic factor *(incorrect -> picks the longest answer)*
>
> *LLaMA2-13B + PackLLM (MedLLaMA2-7B)*: D. All the above *(correct -> identifies all correct answers)*
>
> - Case B
>
> *Input*: "Contributing your $3,000 to a 401(k) or other "
>
> *OPT-6.7B*: “retirement plan today will get you a return of 12-14%”  *(hallucination: incorrect return percentage)*
>
> *OPT-6.7B + PackLLM (OPT-1.3B tuned model on C4)*: “retirement plan offers some tax advantages, but it’s important to take action as early as possible.” *(helpful advice)*

---

> > ### Comment · Reviewer_MKGz · 2024-06-03
> >
> > Thanks for the authors' responses. However, it is still unconvincing to me. For example, what's the point of including cBTM and DExperts in Table 1 without reproducing the full experiments on all compared domain datasets? If all the experts and LLMs are known to cBTM and DExperts methods, do the proposed methods still outperform them? For Table 2, the last "Best Overall" column is misleading; the results are not comparable among the methods. I am still concerned about this.

---

> > > ### Author Response · Authors · 2024-06-04
> > >
> > > **Comparison with cBTM & DExperts**. Thank you for your follow-up questions. cBTM’s LLMs are fine-tuned on top of the OPT model family, which are not relatively powerful models. For example, if we use cBTM’s LLMs, they perform ~25% at accuracy on MMLU, which is close to random answer selection. Generalizing cBTM to other model families for producing full results in Table 1 requires (i) proper data selection, and (ii) further model fine-tuning, which is limiting regarding compute resources. If all experts are known, PackLLM outperforms cBTM (Misc in Table 1)  and DExperts (Medicine in Table 1).
> > >
> > > Following, we highlight the key difference between cBTM, DExperts, and PackLLM (ours) regarding how they compute the importance weights:
> > > - *cBTM*: Importance weights are **data-dependent** (Eq. 8; using tf-idf embeddings); the weights would be similar for LLaMA-7B and LLaMA-70B as the two models have similar pretraining data.
> > > - *DExperts*: Importance weights are **fixed** (Eq. 9).
> > > - *PackLLM*: Importance weights are **model-dependent** (perplexity-based), capturing how well each model understands the prompt; LLaMA-70B would be assigned a higher weight than LLaMA-7B.
> > >
> > > Figure 3 verifies the benefit of model-dependent weights (PackLLM vs. cBTM / DExperts).
> > >
> > > **Table 2**. Thank you for pointing this out. We will remove the “Overall” column to improve the readability of Table 2.

---

### Official Review · Reviewer_7QCH · 2024-05-11

**Rating:** 7
**Confidence:** 3
**Ethics Flag:** 1

**Summary:**

This paper proposes PackLLMs, which utilizes the findings that perplexity of input tokens is a good indicator of LLM's performance in domains. The authors propose PackLLM-simp that utilize the heuristics above and PackLLM-Opt, which approximates an optimization objective with a greedy search algorithm. Experiments show improvement over baselines such as cBTM, DExperts. The approach is further  modular and easy to use compared to other fusion methods that requires training LLMs.

**Questions To Authors:**

I think the paper is quite solid and has addressed my initial questions in their later sections.

A minor comment is that the second paragraph of the related works is repetitive given the second paragraph in introduction.

**Reasons To Accept:**

- The proposed approach is simple and effective, and is broadly applicable to any combinations of LLMs with either same or different architectures.
- The choice of baselines are recent and comprehensive. The results also show clear improvements of LLM.
- The analysis of coefficients obtained (\lambda), and relationships between perplexity and prompt length in Section 6 are quite helpful.
- The writing is quite clear.

**Reasons To Reject:**

- The authors should include discussions about the computational overhead of the approach. How much overhead is introduced by the grid search part, given the high dimensionality of logits (sequence length * vocab size for a single instance)? Is this overhead significant compared to a forward pass of the LM? I also encourage the authors to add a limitations sections to discuss potential limitations of the approach.

---

> ### Author Rebuttal · Authors · 2024-05-31
>
> Thank you for your review. We address your suggestions below.
>
> **Time Complexity**. We provide a time complexity analysis in Table 3, which shows that PackLLM adds negligible time overhead. We also provide the following time measurements for K=5 LLMs per input on a A100 GPU:
>
>
> | |Time|
> |-|-|
> Single forward pass | 0.06 secs|
> |K=5 forward passes (Ensemble)| 0.30 secs|
> PackLLM (K=5) | 0.32 secs |
>
>
> As evaluated, PackLLM does incur significant time costs.
>
>
> **Limitations**. Thank you for your suggestion. We summarize two limitations of our approach below.
>
> (i) One challenge is discussed in Section 3.4 (Tokenizer Selection & Alignment): Equation 1 assumes that the logit vectors $s_k$ can be added together, which requires that the seed LLMs share the same vocabulary. However, this is not always the case, as different LLMs are trained with different vocabularies. While we follow the MinED approach for tokenizer alignment, aligning different tokenizers is an open problem in NLP, and more effective alignment techniques may further improve PackLLM.
>
> (ii) Furthermore, PackLLM incurs additional inference costs compared to model merging approaches, such as weight averaging.
> We will expand on this discussion and include it in a “Limitations” section in the final version.
>
> **Related Work**. Thank you for your suggestion. Upon re-reading, we agree that the related work section is repetitive, we have fixed it in the final version.

---

### Official Review · Reviewer_Gnd4 · 2024-05-16

**Rating:** 7
**Confidence:** 3
**Ethics Flag:** 1

**Summary:**

This paper introduces a new method, PackLLM, to fuse LLMs at test-time as weighted ensemble, where the importance weights for seed LLMs are determined by minimizing the perplexity of the input prompt.

There are two variants of PackLLM, one that directly uses normalized perplexity scores of individual LLMs as weights, and the other that performs an efficient greedy algorithm to search the optimal scores that minimize the prompt perplexity of the ensemble.

The paper conducts extensive experiments to validate the performance of PackLLM. On pretraining perplexity, both variants of PackLLM outperform existing test-time model ensemble baselines like cBTM and DExperts, especially when the number of seed LLMs increases. On downstream tasks, PackLLM also outperforms simple baselines like uniform ensemble. While it trails other learning-based fusion methods like FuseLLM and FoE with the same set of seed LLMs, the learning-free nature allows PackLLM to achieve better performance by swapping in stronger seed LLMs.

The paper also includes some analysis of the weight values, the impact of prompt length, and the time complexity of the two variants.

**Questions To Authors:**

- I’m curious to know the impact of aligning different tokenizers. Have you done any analysis comparing the performance when fusing similar-sized models with the same vs. different vocabularies?
- In experimental results, I’d like to also see a comparison with the best seed LLM performance on each task, especially for Table 4, where most of the baseline results are not available.
- While not an exhaustive search, the greedy search algorithm still seems a very expensive operation that needs to be done for every test-time input. Can we situate the time complexity analysis in Table 3 by comparing the additional time cost from greedy search with the actual inference time of LLMs? This would be important to help understand the real-world applicability of the proposed method.

**Reasons To Accept:**

- Proposes a simple yet effective test-time fusion method for LLMs
- Comprehensive empirical study by comparing PackLLM with different baselines across various settings
- Well-written, clearly articulated motivation

**Reasons To Reject:**

I don’t have major concerns. I do have some questions listed below.

---

> ### Author Rebuttal · Authors · 2024-05-28
>
> Dear Reviewer Gnd4, this review does not correspond to our paper “Pack of LLMs: Model Fusion at Test-Time via Perplexity Optimization”.

---

> > ### Author Response · Authors · 2024-06-02
> >
> > Dear Reviewer Gnd4, thank you for **updating your review** in a timely manner. Please discard our previous message (we are unable to delete it). We reply to your questions below.
> >
> > **Q1) Impact of aligning different tokenizers**. Thank you for your insightful question. We do not have any experimental analysis evaluating the impact of the vocabulary size, but we agree that it would be easier to fuse knowledge from models with similar-sized vocabularies (e.g., LLama + Mistral) than models with different vocabularies (e.g., LLama + Gemma). Disentangling the effects of combining models with different vocabularies, e.g. LLama + Mistral vs. LLama + Gemma, is challenging as different models (Mistral vs. Gemma) perform differently on the underlying tasks.
> >
> > **Q2) Evaluating best LLM**.
> > Thank you for your suggestion. We experiment with the best LLM (as evaluated on test data) on STEM and Commonsense tasks. We find that the best LLM works better when having few LLMs with different vocabularies, but when we have a large number of LLMs with the same vocabulary, PackLLM outperforms the best LLM. We hope with better tokenization mapping methods, PackLLM becomes more effective in the first scenario.
> >
> > To further verify our conclusions, we experiment on the C4 and S2ORC language modeling tasks, where we have a large number of expert OPT models sharing the same vocabulary. In this case, PackLLM also outperforms the Best LLM (selected as the top1 expert).
> >
> > We provide the results in the following tables.
> >
> > Setup: 5 MMLU-STEM tasks and 4 Commonsense tasks.
> > Best LLM: LLaMA2-7B (Fuse LLMs), Platypus-7B (FoE LLMs)
> >
> > | |Fuse LLMs (3 seed LLMs, different vocab) | FoE LLMs (15 seed LLMs, same vocab) |
> > |-|-|-|
> > | | STEM / Com. | STEM / Com. |
> > PackLLM | 31.2 / 54.9 | **40.3** / **67.1** |
> > Best LLM (using test data) | **33.0** / **56.3** | 34.8 / 59.1 |
> >
> > Setup: 9 OPT models (30B OPT + 8* 1.3 OPT experts) on C4 and S2ORC language modeling tasks.
> > Best LLM: Top1 model selection.
> >
> > | Fusion | Perplexity (the lower the better) |
> > |-|-|
> > | | C4 / S2ORC |
> > | Largest (30B) | 12.71 / 11.46 |
> > | Select Best (Top1 model) | 12.53 / 11.46 |
> > |PackLLM | **11.03** / **11.05** |
> >
> > **Q3) Timing of PackLLM**. Thank you for your clarifying question. PackLLM can be applied in real-world scenarios as it does not incur significant additional costs. PackLLM computes the output logits by each LLM once (as in the Ensemble baseline), which are then used for determining the importance weights as shown in Lines 10-15 of Algorithm 1 in Appendix A. No additional LLM forward passes are required.
> >
> > We also provide the following time measurements for K=5 LLMs per input on a A100 GPU:
> >
> > | | Time|
> > |-|-|
> > |Single forward pass | 0.06 secs |
> > | K=5 forward passes (Ensemble) | 0.30 secs |
> > | PackLLM (K=5) | 0.32 secs |
> > As evaluated, PackLLM does not incur significant time costs.

---

> > > ### Comment · Reviewer_Gnd4 · 2024-06-05
> > >
> > > Thank you for the detailed response. It addressed most of my concerns.

---

### Decision · Program_Chairs · 2024-07-10

**Decision:**

Accept

**Comment:**

This work introduces a method named PackLLM to better perform test-time fusion of LLMs as a weighted ensemble as compared to a simple ensemble. This assumes that minimising the perplexity of the combination of LLMs with respect to the input prompt will produce the best model.

Pros:
The work proposes a simple and effective test-time approach for this. The work is also clearly written and clear with results on multiple datasets. The work also has a relatively low test-time computational cost.

Cons:
It would be good to more clearly clarify the difference between alternative baselines including cBTM and DExperts as well as perform thorough comparisons against them where possible.

Despite this, the low test-time computational cost, the improvements in downstream tasks and good performance compared to other baselines such as a simple ensemble and selecting the largest model point to this work likely having a significant impact on the field.